# Gastric Cancer Due to Chronic *H. pylori* Infection: What We Know and Where We Are Going

**DOI:** 10.3390/diseases10030057

**Published:** 2022-08-25

**Authors:** Patrick Joseph Tempera, Mark Michael, Omar Tageldin, Stephen Hasak

**Affiliations:** 1Department of Internal Medicine, Albany Medical Center, Albany, NY 12208, USA; 2Department of Gastroenterology, Albany Medical Center, Albany, NY 12208, USA

**Keywords:** *Helicobacter pylori*, carcinogenesis, gastric cancer, therapeutic advancements, novel therapies

## Abstract

*Helicobacter pylori* is an established cause of many gastrointestinal pathologies including peptic ulcer disease, gastritis, and gastric cancer. It is an entity that affects the global population, and its true nature has only been known since the 1980s. Although there is much known about *H. pylori* including its pathophysiology, detection, and eradication, resistance to current therapy models is common. This is problematic because untreated or inadequately treated *H. pylori* increases morbidity and mortality related to gastric cancer and peptic ulcer disease among others. In order to improve the treatment and reduce resistance, there is significant ongoing research identifying new detection and eradication methods for *H. pylori*. This review aims to highlight what has already been established regarding *H. pylori*’s epidemiology, pathophysiology, detection, and treatment as well as the most current and novel research involving detection and treatment of *H. pylori*.

## 1. Introduction

*Helicobacter pylori* is a bacterium that is commonly found amongst the generally healthy population. However, it has also been found in patients with gastric pathologies such as chronic gastritis, peptic ulcer disease (PUD), and gastric cancer. It was first reported by Bottcher and Letulle in 1875 when they observed the bacteria present on the margins of peptic ulcers [1]. However, the true nature of the bacterium was not discovered until the 1980s by B.J. Marshall when he isolated and cultured samples from peptic ulcer disease and identified and described causation between PUD pathology and *H. pylori* [2]. Since that time, many gastrointestinal diseases have been paired to the existence of *H. pylori* in the gastric environment: gastric and duodenal ulcers in 1 to 10% of infected patients, gastric carcinoma in 0.1 to 0.3%, and gastric mucosa associated lymphoid tissue (MALT) lymphoma in less than 0.01% [3]. *H. pylori* is thought to be introduced to the gastric environment during the early years of life and persist indefinitely unless treated. Although present from a young age, it can remain silent and undiagnosed until gastric pathology manifests due to chronic inflammation of the underlying gastric mucosa. *H. pylori* possesses many virulence factors that allow it to survive the toxic and harsh environment of the stomach and change the surrounding environment such that it promotes carcinogenesis ultimately leading to gastric cancer. Once diagnosed with gastric cancer, the prognosis is poor due to discovery at an advanced age and advanced disease state [4]. Importantly, eradication of *H. pylori* can prevent gastric cancer from forming. It is imperative that we understand our current treatment models as well as novel therapeutic agents targeted for *H. pylori* eradication. This review revisits established epidemiology of and treatment for *H. pylori* and looks further into the future regarding novel therapeutic potentials.

The prevalence of *H. pylori* is high in many countries with several studies reporting that 50% of the world’s population is infected. The prevalence ranges from 20% to 50% in industrialized countries to over 80% in developing countries [5]. Within developing countries, most children are infected before the age of 10 and by the age of 50, within the same region, more than 80% of the population have been infected [5]. In northern Europe and North American populations, about one-third of adults are infected, with south and east Europe, South America, and Asia having an *H. pylori* prevalence often higher than 50% [4]. However, with the introduction of awareness and better diagnostic and therapeutic modalities, the epidemiology of *H. pylori* infection has been changing with a noticeable decline in prevalence in most countries over the last few decades. Additionally, primary prevention of *H. pylori* is a major topic of research with further investigation given towards identifying risk factors that can be avoided. It has been shown that gender and age are not associated with an increase in risk of infection with most studies reporting that there is no significant difference between men and women, both in adults and in children [4]. Factors that have been shown to increase the chances of an *H. pylori* infection include low socioeconomic conditions in childhood, living in rural areas, crowded homes, and having contaminated water sources [4]. Individuals who smoke and drink were also found to have elevated risk. Having identified some of the conditions that increase one’s likelihood of having an *H. pylori* infection, the exact route of transmission is still not known. It is believed that infection most likely occurs with person-to-person transmission which is subdivided into two potential mechanisms: vertical and horizontal [2]. Vertical transmission involves the contraction of *H. pylori* within the confines of one family, while horizontal transmission of *H. pylori* describes transmission to an individual due to contact with individuals and/or environmental objects outside of the individual’s family. [2]. Additionally, *H. pylori* has been detected in the oropharyngeal spaces: dental plaque, saliva, tonsil tissue, root canals, and oral mucosa. Many authors would agree that intrafamilial transmission of *H. pylori* is likely the dominant method and most significant route [2]. Regardless of socioeconomic status or route of transmission, once infected with *H. pylori* and without proper diagnosis and treatment, chronic infection will increase the risk of developing gastric cancer 1.4 to 4.2 times over the risk in the non-infected general population.

## 2. Review of Pathophysiology/Pathogenesis

*H. pylori* is recognized as a grade 1 carcinogen by the International Agency for Research on Cancer [6]. *H. pylori* has been linked to the development of chronic active gastritis and atrophic gastritis which can lead, albeit over time, to cancer through a sequence of cellular events. There have been two randomized controlled trials with the development of gastric cancer as the primary outcome: Wong et al. and Fukase et al. [7,8]. Both studies were performed amongst Far East populations at an increased risk for *H. pylori* infection [5]. Wong et al. followed 1630 *H. pylori* positive patients for 7 years. The group of patients with no eradication measures taken had 11/813 (1.4%) develop gastric carcinoma compared to 7/817 (0.9%) patients that received eradication therapy developing gastric carcinoma [7]. Fukase et al., similarly, enrolled 544 patients located in Japan with a 3-year follow-up. 9/272 (3.3%) patients that received *H. pylori* eradication therapy developed gastric cancer and 24/272 (8.8%) patients that did not receive therapeutic intervention developed gastric cancer [8]. Both studies highlight the clear association between *H. pylori* infection and the development of gastric adenocarcinoma. For this to take place, *H. pylori* must be present for a prolonged period to create an environment conducive for itself and eventually, carcinogenesis. *H. pylori* possesses many virulence factors and properties that allow it to survive in the harsh gastric environment in addition to causing chronic inflammation and eventual carcinogenesis.

*H. pylori* pathogenesis and sequential outcomes are accomplished by a combination of virulence factors that are dependent on both host and environmental factors. Pathogenesis first requires *H. pylori* entry into the stomach after which colonization and infection result in disease outcomes. First, *H. pylori* must survive the acidity of the stomach, move toward the gastric epithelium via flagella, and attach to host cells using adhesins. Receptor interaction then allows bacteria to cause damage to tissues by toxin release [9]. The most favorable conditions for *H. pylori* are 37 °C, pH of 4.0 to 6.0, microaerophilic conditions, and adequate water and nutrients [2]. Once *H. pylori* enters the stomach, it can alkalize its surrounding environment by using the innate enzyme urease, encoded by the urease gene cluster to neutralize hydrochloric acid [9]. Once the surrounding environment is a viable pH, the bacteria move towards the host gastric epithelium using flagella-mediated motility. The favored location for *H. pylori* is in the first part of the duodenum within the super-epithelial mucus layer gastric pits. It is not detected in the subepithelial space of the stomach or in the epithelium of the gastric glands [2]. Once it arrives at the appropriate layer of host epithelium, *H. pylori* uses specific adhesins to attach to host cell receptors allowing for colonization and persistent infection. After *H. pylori* is established within the correct position, it will release virulence factors such as cytotoxin-associated gene A (CagA) and vacuolating cytotoxin A (VacA). With the release of these virulence factors, tissue damage occurs, and the body responds by secreting chemokines that initiates an immune response furthering inflammation [9].

Shown using mouse models, after CagA translocates into host epithelial cells, it enhances mitotic activity of gastric epithelial cells contributing to malignancy if left unchecked [9]. Several studies indicate that *H. pylori* strains with CagA are directly associated with acute gastritis, gastric ulcer, and gastric cancer development [10,11]. Unlike CagA, which has one main virulence process, VacA possesses many. After internalization, VacA creates large pores in the cytoplasmic membrane, allowing for further bacterial urease effect on the gastric epithelial cell [9]. Eventually, VacA enters the epithelial cell mitochondria targeting pro-apoptotic factors promoting increased epithelial cell-turnover. Additionally, VacA can control host immune T-lymphocytes by reducing their activation and decreasing the host’s inflammatory response allowing for *H. pylori*’s continued survival [9]. Combining the innate survival techniques and toxin producing abilities that *H. pylori* possesses, continued infection and inflammatory process will eventually allow for carcinogenesis to take place. 

## 3. Diagnostic Methods in Practice

To appropriately diagnose *H. pylori*, there are several tests clinically available. These tests can be classified into two groups: invasive testing and non-invasive testing. Invasive testing requires endoscopic evaluation for tissue sampling allowing for histology, rapid urease test, culture, and molecular methods. Non-invasive testing includes urea breath test, stool-antigen testing, and serological testing.

Invasive testing has the benefit of providing direct visual evidence of infection such as gastritis or ulceration, but these are nonspecific in most cases. It can also diagnose advanced pathology such as cancer. After obtaining a biopsy via endoscopy, the specimen can be used for direct detection of *H. pylori* infection. However, the diagnostic accuracy of this method depends on the biopsy site, size, number of biopsies, staining methods used, use of proton pump inhibitors (PPIs), recent use of antibiotics, and experience of pathologists [12]. To counteract some of these modifiable risks, it is recommended that two weeks prior to histologic testing, all PPIs and antibiotics be stopped. Samples should be obtained from standardized sites: the antrum and corpus of the stomach, with two biopsies taken from each section to minimize sample error and increase the odds of accurate diagnosing. An additional test that can be used after biopsy is the rapid urease test (RUT). After a sample is obtained and prepared, a urea antigen agent is added to measure the activity of the *H. pylori* urease enzyme by converting to ammonia and leading to an increase in the sample pH and a color change on pH monitor paper. Use of H2-receptor antagonists, PPIs, antibiotics, bismuth compounds, and the presence of blood reduce the accuracy of RUT. However, this method is inexpensive, easy to perform, specific, provides results in a timely manner, and widely available [13].

Tissue culture is another method that can be used for *H. pylori* testing. Culturing for *H. pylori* is time-consuming and expensive for laboratories and there are many factors that may interfere with accurate results: poor quality of specimens, delayed transport, exposure to aerobic environment during transport and preparation, and inappropriate mediums required for growth [14]. However, culturing is extremely useful because it can also help determine antibiotic sensitivity for strains. Lastly, polymerase chain reaction (PCR) can be used for the diagnosis of *H. pylori*. PCR amplifies *H. pylori* genetic material from saliva, stool, and gastric juice samples making it favorable because it requires fewer bacteria in samples, faster results, and no need for special processing supplies or transportation like that of culturing [15]. PCR will also detect mutations in *H. pylori* that may indicate antibiotic resistance allowing for more appropriate treatment regimens to be used.

Non-invasive tests include the urea-breath test, stool antigen test, and the antibody-based tests. The most popular of these is the urea breath test, which has been used in practice for close to 30 years. It is considered the most accurate non-invasive test for diagnosis of *H. pylori* infection. It detects the urease activity of *H. pylori* using carbon labeled urea that is ingested by patients and hydrolyzed to labeled CO_2_ in the stomach if *H. pylori* is present [12]. The labeled CO_2_ is absorbed in the blood and exhaled which is then measured. Much like invasive testing, there are conditions in which the urea breath test will have decreased diagnostic accuracy. PPIs, antibiotics, and active bleeding can lead to false negative results. However, in the absence of these, the urea breath test has a 95% sensitivity and specificity [16]. The stool antigen test can detect the presence of *H. pylori* in stool samples by measuring either enzyme immunoassay or immunochromatography assay. These tests can also be used for epidemiological studies and screening programs [17]. Lastly, antibody-based testing is another non-invasive test for *H. pylori*. Antibody-based testing detects anti-*H. pylori* IgG antibodies in a patient’s serum. However, it cannot distinguish between active and past exposure to *H. pylori*, so this cannot confirm active infection or confirm eradication after treatment; one of the advantages of this test is that it is not influenced by PPIs or antibiotic use [18].

## 4. Treatment Regimens in Practice

Current treatment modalities for successful eradication of diagnosed *H. pylori* rely on antimicrobial agents and antisecretory agents. Antimicrobial agents used for *H. pylori* treatment include but are not limited to clarithromycin, levofloxacin, metronidazole, tetracycline, rifabutin, and bismuth-containing compounds [19]. Used synergistically to achieve antibiotic bactericidal effect, antisecretory medications like H2-receptor antagonists and PPIs are also used. PPIs are mostly preferred used because they are most effective in increasing the gastric pH when compared to that of H2-receptor antagonists. PPIs have also been shown to impose antimicrobial activity against *H. pylori* [20].

To best combat *H. pylori* infection, many combination therapies using antibiotics and antisecretory agents have been studied. The Maastricht I Consensus Report recommends that treatment regimens used should achieve an eradication rate of at least 80% and propose a standardized report card to be used to evaluate the outcome of new therapeutic regimens for *H. pylori* infection [21]. Furthermore, specific guidelines are put forth for regions and populations depending on *H. pylori* infection rate, prevalence specific to a particular area, and resistance patterns. Despite the different studied treatment regimens and region-tailored treatments, guidelines for first line and rescue therapies are generally similar. Traditionally, the standard first-line therapy contains a PPI and two antibiotics, usually clarithromycin and amoxicillin. Metronidazole can substitute amoxicillin if there is a penicillin allergy. The recommended therapeutic duration of standard therapy is 10 to 14 days in the United States and 7 days in Europe and Asia. If first line therapy were to fail and no proof of eradication was obtained, bismuth-containing quadruple therapy or levofloxacin based triple therapy are recommended as a rescue therapies [19]. However, new data has emerged illustrating changes in *H. pylori* antibiotic susceptibility. A meta-analysis of studies comparing clarithromycin triple therapy and bismuth quadruple therapies performed from around the world suggest that these two treatments are similar in efficacy, compliance, and tolerability [22]. Furthermore, a recent meta-analysis including 12 random controlled trials and 2753 patients showed eradication rates of 77.6% with bismuth quadruple therapy compared to 68.9% eradication rate with clarithromycin triple therapy with ten days of bismuth quadruple therapy found to be more effective than 7 days of clarithromycin therapy [23]. The most recent meta-analysis of *H. pylori* eradication treatments is continuing to show that 10 to 14 days of bismuth quadruple therapy is superior to 7 days of clarithromycin triple therapy with an 85% compared to 73% eradication rate, respectively [24]. Based on these data, it can be proposed that a 10-to-14-day course of bismuth quadruple therapy is recommended over clarithromycin triple therapy.

The importance of *H. pylori* eradication is illustrated by Choi et al. in which a randomized control trial of 1838 individuals with diagnosed *H. pylori* and confirmed first-degree relatives with gastric cancer were analyzed after eradication treatment. This group was randomly assigned to receive eradication therapy or placebo and followed for 9.2 years. Over this time, 10 participants in the treatment group and 23 participants in the placebo group developed gastric cancer. Interestingly, of the 10 participants in the treatment group in whom gastric cancer developed, 5 had persistent *H. pylori* infection [25]. Ultimately, this study highlights that even amongst patients with a first-degree familial history of gastric cancer, eradication of *H. pylori* reduced their risk of gastric cancer [25].

Additionally, Lee et al. performed a systemic review and meta-analysis of randomized controlled trials and observational studies to investigate the effects of *H. pylori* eradication on the incidence of gastric cancer and found that individuals with eradication of *H. pylori* infection had a lower incidence of gastric cancer than those who did not receive eradication therapy [26]. This study further found that regardless of a patient’s baseline incidence of gastric cancer, the eradication of *H. pylori* still benefited by lowering the overall incidence of gastric cancer.

More recent data has been showing that the eradication rate of first-line therapy is decreasing to less than 80% in some areas [19]. Having established that the chronicity of *H. pylori* infection can lead to gastric cancer and current therapeutic models are seeing diminished eradication rates, research into alternative and novel therapeutic options is necessary to ensure that eradication of *H. pylori* is still attainable in the years to come.

## 5. Advancements in Therapy and/or Diagnostic Modalities

Recent studies have taken interest in determining the risk of developing gastric cancer secondary to *H. pylori*, given that current testing methods are relatively affordable and accurate. One study assessed the relationship between *H. pylori* virulence factors and the expression of immune checkpoint inhibitors as well as Th1, Th17, and Treg response genes [27]. Virulence factors included cytotoxin associated gene A (CagA) and vacuolating cytotoxin gene A (VacA); immune checkpoint inhibitors included PD-1 and PD-L1; and response genes included interferon gamma, IL-17, and FOXP3. This study found that elevated PD-1:PD-L1 inhibitor levels act as a diagnostic predictor for gastric cancer as elevated levels were found in precancerous lesions [27].

Another study looked for serum biomarkers associated with *H. pylori* gastritis, which places patients at a higher risk of developing gastric cancer [28]. They found the ratio of pepsinogen 1 to pepsinogen 2 (PG1/PG2) was the most reliable test to detect chronic atrophic gastritis compared to PG1, PG2, gastrin 17 or anti-*H. pylori* antibodies, given that oxyntic cell atrophy in *H. pylori* infection leads to a more significant PG1 reduction [28]. The authors determined that the PG1/PG2 ratio has a specificity of 83% (95% CI: 0.64–0.93) in detecting chronic atrophic gastritis, although this test was notably not effective at determining low or high-risk populations [28]. 

Various other promising targets have been evaluated in recent studies regarding reduction of gastric cancer or improvement in treating *H. pylori* infection. Courtois et al. assessed CD44 (a gastric cancer marker) and LC3 (autophagy marker) expression in the setting of *H. pylori* infection. They found that when autophagy inhibitors were used to treat infected cells, migration ability and CD44 expression were reduced [29]. The authors posit that this relationship can be targeted with a treatment to reduce *H. pylori* induced autophagy thereby decreasing inflammation and gastric cancer risk.

Given the emerging resistance of *H. pylori* to current therapies, alternative treatment targets must be identified. Two such targets are heparanase and carbonic anhydrase. Tang et al. report *H. pylori* infections up-regulate heparanase levels, which allows for easier colonization, as well as recruitment of immune cells, resulting in worsening inflammation and gastritis [30]. Heparanase acts by breaking up heparin sulfate, resulting in extracellular matrix remodeling and easier extravasation of cytokines and immune cells [30]. Another treatment target is carbonic anhydrase with the goal of reducing biofilm production and resistance to antibiotics [31]. Grande et al. trialed two carbonic anhydrase inhibitors (CAI), carvacrol and thymol. They found that CAI intervention reduced growth of *H. Pylori*, biofilm production, and are more isoselective compared to acetazolamide. They concluded that carvacrol and thymol may be useful in multi-drug therapy to treat infections and reduce antibiotic resistance with less gut microbiome effect compared to amoxicillin.

In terms of antibiotics and resistance, Haseena et al. assessed amoxicillin and adjuncts that can enhance treatment. They developed a novel drug carrier to improve amoxicillin activity by using a lactobionic acid coated zinc metal-organic framework (LBA coated Zn-MOF). They found that the half maximal inhibitory concentration, minimum inhibitory concentration, and minimum biofilm inhibitory concentration of amoxicillin was reduced by nearly 10-fold when used with a LBA coated Zn-MOF [32].

Given PPI use and the importance of managing gastric pH, a novel drug has recently been approved in China and is gaining popularity; Vonoprazan. This is a reversible H+K+ ATPase inhibitor that has a faster onset and longer half-life than current PPIs [33]. Chan et al. conducted a meta-analysis of RCTs involving Vonoprazan and they determined that treatment involving Vonoprazan had higher eradication rates of 87–90% compared to PPI therapy with eradication rates of 75.4–79%, 95% CI: 1.04–1.23, *p* < 0.003 [33].

Another target studied by Zahra et al. is to inhibit *H. pylori*’s urease with a novel antibiotic. Urease produces ammonia and is instrumental for colonization of the GI system [34]. The authors identified acetylphenol-based acylthioureas, most notably Drug 7f. They found that Drug 7f had a ~413-fold inhibition potential compared to the positive control thiourea, with an IC50 of 0.054 uM [34].

It is important to mention some of the ongoing clinical trials in both the detection and treatment of *H. pylori*. In one trial, *Helicobacter Pylori* Genome Project, Rabkin et al. are researching the genetic and epigenetic variations in *H. pylori* strains among chronically infected individuals [35]. The hope of this study is to add an additional layer of *H. pylori* detection to understand if a particular genetic/epigenomic finding gives insight into the risk of progression to gastric cancer. This study may help characterize virulence among *H. pylori* strains and provide biomarkers for early diagnosis of gastric cancer [35].

In additional to genetic testing, advancements in endoscopy are also under evaluation. Robles-Medranda et al. are testing a new technology that combines high-definition imaging with optical magnification with the ability to increase images up to 136 times with a better quality of image than that of standard scopes [36]. This will optimize the gross evaluation of mucosa and superficial vascular areas to better identify early signs of inflammation or lesions not seen before while using standard endoscopy. An additional technologic advancement focuses on making it easier to identify slight differences in color of the mucosa facilitating the detection of *H. pylori* infection and gastric atrophy by using Linked Color Imaging (LCI) [37]. LCI enhances differences in hue by making red areas appear redder and white areas appear whiter to help better differentiate between normal and abnormal mucosa.

Lastly, the effects of radioactive iodine on eradication of *H. pylori* in patients treated by thyroid disease is being evaluated. Six to eight weeks and 6 months after administration of radioactive iodine (131), patients with thyroid disease and *H. pylori* will undergo stool antigen testing to assess for the eradication of *H. pylori* [38]. It is not yet known the exact mechanism of action, but it is hypothesized that iodine (131) promotes *H. pylori* toxicity.

If these developments are found to be clinically viable, clinicians have more tools than ever before when managing *H. pylori* infections. As further advancements are made in the field, resistance to antibiotic treatment may be overcome and cancer development may be reduced.

## 6. Conclusions

This review summarizes *H. pylori* literature regarding epidemiology, pathophysiology/pathogenesis, and diagnostic and therapeutic methods currently used in medical practice. The understanding of risk factors and pathogenesis of *H. pylori* is of great importance since untreated infection supports a chronic inflammation environment promoting carcinogenesis. The association between chronic inflammation and cancer development is widely accepted, and this review highlights some of *H. pylori*’s notorious virulence factors by which chronic infection is preserved. Diagnostic methods have been developed to identify active *H. pylori* infection and similarly, we possess the ability to treat *H. pylori*. However, identification and treatment continue to remain inadequate with increasing incidence rates of *H. pylori* and treatment resistance. Advancements in both diagnosis and treatment are necessary to ensure that *H. pylori* can be checked in its disease progression, carcinogenesis, and incidence of gastric cancer. A summary of these advancements in both identification and treatment can be seen in Table 1 and Table 2 respectively.

## Figures and Tables

**Table 1 diseases-10-00057-t001:** Novel diagnostic modalities summarized.

Diagnostic Advancements	Action	Study
PD-1:PD-L1	Diagnostic predictor for gastric cancer; elevated levels were found in precancerous gastric lesions.	27
Pepsinogen 1:Pepsinogen 2	Serum biomarker for *H. pylori* gastritis.	28
*Helicobacter Pylori* Genome Project	*H. pylori* genetic/epigenomic biomarker for early gastric cancer diagnosis.	35
Analyzing gastric mucosa	Magnifying images 136 times with better quality to assess early signs of inflammation.	36
Linked Color Imaging	Helps better differentiate between inflamed mucosa from normal mucosa.	37

**Table 2 diseases-10-00057-t002:** Novel therapeutic modalities summarized.

Therapeutic Advancements	Action	Study
LC3 (autophagy marker) CD44 (gastric cancer marker)	Potential therapeutic targets. Autophagy inhibitors decreased *H. pylori* migration and CD44 expression.	29
Heparanase	Found to be upregulated in *H. pylori* infection. When blocked, *H. pylori* colonization and immune cell recruitment decreased.	30
Carbonic Anhydrase	When blocked, decreased *H. pylori* biofilm ability and antibiotic resistance.	30,31
Novel Drug Carriers	Improves amoxicillin activity.	32
Vonoprazan	Reversible H+K+ ATPase inhibitor with faster onset and longer half-life than current PPIs.	33
Novel Antibiotic Drug 7f	Inhibits *H. pylori*’s urease.	34
Radioactive Iodine (131)	Toxic to *H. pylori*.	38

## Data Availability

Not applicable.

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
