# Peer review of "Gastric Cancer Due to Chronic H. pylori Infection: What We Know and Where We Are Going"

_diseases, 2022, doi:10.3390/diseases10030057_

Round 1

Reviewer 1 Report

Helicobacter pylori-related diseases are among the most prevalent in the world emphasizing the need for all physicians to be knowledgeable regarding the infection, its diagnosis and management.

The review by Tempera et al aimed “to highlight what has already been established regarding H. pylori’s epidemiology, pathophysiology, detection, and treatment as well as the most current and novel research involving detection and treatment of H. pylori

The purpose was interesting however the manuscript did not reach the goal! Much more updated papers on this topic are present at the moment in the literature

First of all the title is to high-sounding and do not reflect the manuscript contents

Several original references need to be quoted.

For example: It was first reported by Bottcher and Letulle in 1875

Marshall BJ, Warren JR. Unidentified curved bacilli in the stomach of patients with gastritis and peptic ulceration. Lancet. 1984 Jun 16;1(8390):1311-5. doi: 10.1016/s0140-6736(84)91816-6. PMID: 6145023.

Etc…………….

The section “Review of Pathophysiology/Pathogenesis” is really too superficial

Moreover, in the section “Current Diagnostic Methods” although is titled “current…………..” there are only 2 very old (2014) references quoted and not the specific references.

In the end the manuscript appears as truncated, without conclusions or summary.

Finally, an extensive English editing is needed

For all these reasons, the manuscript is not suitable for publication

Reviewer 2 Report

In this paper, the authors reviewed the epidemiology, pathophysiology, detection, and treatment of Helicobacter pylori (H. pylori) infection.  As they highlighted, H. pylori is a gastric pathogen that colonizes approximately 50% of the world's population. Notably, in 2005, Marshall and Warren were awarded the Nobel Prize in Medicine for their seminal discovery of this bacterium and its role in peptic ulcer disease.

Based on current international guidelines and a network meta-analysis comparing the effects of various treatment regimens, nonbismuth quadruple therapies for 10-14 days and vonoprazan-based triple therapy for 7 days are the currently recommended H. pylori treatment regimens. 

Overall, I have a positive opinion of the article. Nevertheless, I suggest revising it to further improve its value.

Major concerns

The role of H. Pylori eradication in cancer prevention should be better discussed and other studies should be cited. For example, in an RCT investigation, Choi et al. (doi: 10.1056/NEJMoa1909666) demonstrated that among persons with H. pylori infection with a family history of gastric cancer in first-degree relatives, bacterial eradication treatment reduced the risk of gastric cancer.  Yet, in a systematic review/metaanalysis, it was proved that the positive effects of eradication depend on several variables such as baseline gastric cancer incidence, but apply to all levels of baseline risk.

Please, take a look at the clinical research on the topic (clinicaltrial.gov) https://clinicaltrials.gov/ct2/results?cond=helicobacter&term=cancer+gastric&cntry=&state=&city=&dist=

as I found several interesting findings

Minor issues

In the table, I would suggest distinguishing the columns in "diagnosis" and "treatment"

Please consider that treatment should also focus on minimizing the potential for increasing antimicrobial resistance or causing gut microbiota dysbiosis.

Reviewer 3 Report

In the manuscript titled "Gastric cancer due to chronic Helicobacter pylori infection: what we know and where we are going" Patrick Tempera and colleagues, reported that H. pylori’s epidemiology, pathophysiology, detection, and treatment as well as the most current and novel research involving detection and treatment of H. pylori. I have a few comments regarding the present manuscript.

-Please change the error in the title. H. pylori with the uppercase for the H, and italics for the name in the entire document.

-Please delete (H. pylori) in the first line of the abstract and introduction.

-Epidemiology and H. pylori infection, please delete the title and add the whole section to the introduction.

-The other sections look well and summarize the main information.

-Dear Editorial office, please check the similarity with other papers published, I know that you have this tool and the reviews have to be less than 30% of similarity.

Author Response

-Please change the error in the title. H. pylori with the uppercase for the H, and italics for the name in the entire document.

- This has been edited to reflect your recommendation. Thank you!

-Please delete (H. pylori) in the first line of the abstract and introduction.

- This has been edited to reflect your recommendation. Thank you!

-Epidemiology and H. pylori infection, please delete the title and add the whole section to the introduction.

– This has been edited to reflect your recommendation. Thank you!

-Dear Editorial office, please check the similarity with other papers published, I know that you have this tool and the reviews have to be less than 30% of similarity.

- We personally use iThenticate. I am unsure which software the Editorial Office uses. Uploading our review and when excluding the cited works, we scored a 26%. I hope this helps when making a final decision. Thank you!

Round 2

Reviewer 1 Report

Although the authors made some efforts to improve their manuscript, the review still remain of a very low quality, it is superficial and add very little to what at the moment is present in the literature. Moreover it is difficult to follow because it requires a deep editing of English language and style and in my opinion the manuscript has to be rejected.

Author Response

Dear Reviewer 1,

Thank you for providing comments and recommendations for our original draft. We are sorry to hear that our revised manuscript is not approved by your standards. We will continue to make strides to improve our content and literary style. I hope this message finds you well.

Sincerely,

Patrick J. Tempera

Reviewer 2 Report

The authors have addressed my original comments satisfactorily. I endorse this publication.

Author Response

Thank you for providing your comments and recommendations for our original draft. They were insightful and provided clear direction that we feel allowed us to improve our content for the ultimate betterment of consolidating what is known about H. pylori and its disease process. We are grateful for your endorsement and we hope we can continue to work together to publish our work in this fine journal.

Sincerely,

Patrick J. Tempera

Reviewer 3 Report

In the abstract, keyword, introduction first line, and page four about anti-H Pylori IgG, there are some errors pertaining to Helicobacter. Regarding the references, they are bold, please change them.
